# Non-Canonical Activin A Signaling Stimulates Context-Dependent and Cellular-Specific Outcomes in CRC to Promote Tumor Cell Migration and Immune Tolerance

**DOI:** 10.3390/cancers15113003

**Published:** 2023-05-31

**Authors:** Mark B. Wiley, Jessica Bauer, Kunaal Mehrotra, Jasmin Zessner-Spitzenberg, Zoe Kolics, Wenxuan Cheng, Karla Castellanos, Michael G. Nash, Xianyong Gui, Lyonell Kone, Ajay V. Maker, Guilin Qiao, Deepti Reddi, David N. Church, Rachel S. Kerr, David J. Kerr, Paul J. Grippo, Barbara Jung

**Affiliations:** 1Department of Medicine, University of Washington, Seattle, WA 98195, USA; mwile2@uw.edu (M.B.W.); kunaal.mehrotra@gmail.com (K.M.);; 2Clinical Department for Gastroenterology and Hepatology, Medical University of Vienna, 1090 Vienna, Austria; 3Department of Medicine, Division of Gastroenterology and Hepatology, University of Illinois at Chicago, Chicago, IL 60607, USA; 4Department of Biostatistics, University of Washington, Seattle, WA 98195, USA; 5Department of Laboratory Medicine and Pathology, University of Washington, Seattle, WA 98195, USA; 6Department of Surgery, University of Illinois at Chicago, Chicago, IL 60607, USA; 7Department of Surgery, University of California-San Francisco, San Francisco, CA 94115, USA; 8Nuffield Department of Medicine, University of Oxford, Oxford OX1 4BH, UK; 9NIHR Oxford Comprehensive Biomedical Research Center, Oxford University Hospitals NHS Foundation Trust, University of Oxford, Oxford OX1 4BH, UK; 10Department of Oncology, University of Oxford, Oxford OX1 4BH, UK; 11Radcliffe Department of Medicine, University of Oxford, Oxford OX1 4BH, UK

**Keywords:** activin A, colorectal cancer, tumor microenvironment, metastasis, digital spatial profiling

## Abstract

**Simple Summary:**

The identification of markers of metastatic disease is critical to improving colorectal cancer survival rates. Activin A plays a critical role in tumor and immune cells in colorectal cancer. Here, we aimed to expand on our previous work to determine how activin A influences tumor cell signaling and immune activation status. Our findings suggest that activin A inhibits the activity of immune cells found in the tumors of colorectal cancer patients while promoting metastasis. Furthermore, we provide data suggesting activin A stimulates a targetable pathway in tumor cells and immune cells leading to metastatic outcomes. This work highlights the application of novel, commercially available imaging techniques to colorectal cancer patient samples to confirm signaling networks driving disease progression. The identification of activin-induced pathways driving metastasis may provide an early target for colorectal cancer to inhibit progression.

**Abstract:**

We have shown that activin A (activin), a TGF-β superfamily member, has pro-metastatic effects in colorectal cancer (CRC). In lung cancer, activin activates pro-metastatic pathways to enhance tumor cell survival and migration while augmenting CD4+ to CD8+ communications to promote cytotoxicity. Here, we hypothesized that activin exerts cell-specific effects in the tumor microenvironment (TME) of CRC to promote anti-tumoral activity of immune cells and the pro-metastatic behavior of tumor cells in a cell-specific and context-dependent manner. We generated an Smad4 epithelial cell specific knockout (*Smad4*−/−) which was crossed with TS4-Cre mice to identify SMAD-specific changes in CRC. We also performed IHC and digital spatial profiling (DSP) of tissue microarrays (TMAs) obtained from 1055 stage II and III CRC patients in the QUASAR 2 clinical trial. We transfected the CRC cells to reduce their activin production and injected them into mice with intermittent tumor measurements to determine how cancer-derived activin alters tumor growth in vivo. In vivo, Smad4−/− mice displayed elevated colonic activin and pAKT expression and increased mortality. IHC analysis of the TMA samples revealed increased activin was required for TGF-β-associated improved outcomes in CRC. DSP analysis identified that activin co-localization in the stroma was coupled with increases in T-cell exhaustion markers, activation markers of antigen presenting cells (APCs), and effectors of the PI3K/AKT pathway. Activin-stimulated PI3K-dependent CRC transwell migration, and the in vivo loss of activin lead to smaller CRC tumors. Taken together, activin is a targetable, highly context-dependent molecule with effects on CRC growth, migration, and TME immune plasticity.

## 1. Introduction

Colorectal cancer (CRC) is the third leading cause of cancer-related deaths in the US and remains a significant healthcare challenge [1]. The screening of asymptomatic, low-risk individuals has been effective in reducing mortality [2]. However, we are now grappling worldwide with a surge in late-stage CRCs in patients aged 20–39 years old [3]. Once cancer is diagnosed, mortality is stage-dependent and remains high at later stages. A better understanding of targetable signaling networks is needed for the advancement of novel therapeutics.

Activin A (activin) is a member of the transforming growth factor-β (TGF-β) superfamily and contributes to CRC development [4]. Activin inhibits the growth of epithelial cells, induces apoptosis [5], and promotes metastasis [6]. The “canonical” activin signaling constitutes ligand interaction with an activin sub-type II (ACVR2A/2B) and type I (ACVR1A/1B/1C) serine-threonine kinase receptor, culminating in the nuclear SMAD4 translocation for transcriptional regulation [6,7]. We have previously shown that activin stimulation leads to the activation of the canonical SMAD pathway in CRC cells [8]; however, several Smad-independent signaling pathways have been identified. These “non-canonical” activin pathways signal via PI3K/AKT, MAPK/ERK, WNT/β-catenin, and notch pathways [6,8] and are associated with cellular growth and differentiation [9,10]. Importantly, our group has reported that CRC cells stimulated with activin displayed increased phosphorylation of AKT which was exacerbated in the absence of SMAD4 in CRC cells [8]. This publication also found that the loss of ACVR2A resulted in loss of pAKT signaling in vivo, suggesting that activin requires this receptor to stimulate the PI3K/AKT pathway [8].

In addition to cancer cells, activin signals along innate and adaptive immune cells to stimulate highly-cell-specific and context-dependent outcomes. Macrophages produce activin, which can stimulate an M1 phenotype of these cells [11]. Macrophages stimulated with activin produce increased mRNA of TNFα and IL-β in vitro [12]. Furthermore, activin increases nitric oxide and IL-1β release and increased the phagocytic activity of mouse peritoneal macrophages, suggesting a pro-inflammatory role for this cytokine in innate immune cells [13]. Activin’s role in adaptive immunity appears to be highly context dependent. Isolated CD4 T-cells from rodents stimulated with activin have been shown to polarize these cells into regulatory T-cells (T_reg_) in vitro, and transgenic mice with elevated serum activin display increased circulating T_regs_ [14]. Opposingly, activin treatment reduced the tumor burden, increased the survival time, and increased the pro-inflammatory profile of isolated CD4^+^ T-cells in a Lewis lung carcinoma (LLC) mouse model of lung cancer [15]. Additionally, several markers of T-cell exhaustion were reduced following systemic activin treatment in this mouse model including FOXP3, PD-1, and CTLA-4 [15]. These data suggest that the model employed and status of the T-cells prior to treatment are critical in determining the effect of activin stimulation.

The state of the peritumoral immune infiltrate is critical in microsatellite instability (MSI), the genomic mechanism in approximately 15% of sporadic CRCs, which is associated with good response to immunotherapy [16]. Conversely, the more common microsatellite stable (MSS) CRCs are less responsive [17]; therefore, the identification of signaling networks which promote an anti-tumoral immune response is critical to enhancing therapeutic strategies. A recent RNA sequencing analysis identified that TGF-β signaling signatures are associated with the ratio of exhausted:resident infiltrating lymphocytes, which is also associated with survival time in CRC [18]. Additionally, a loss of TGF-β in the epithelial cells in CRC leads to a non-specific inflammatory response that promotes tumor progression [19], suggesting a significant role for TGF-β in directing an appropriate immune response in the TME. There is significant cross-talk between activin and TGF-β in CRC, with several described effects of TGF-β requiring simultaneous activin stimulation [4]. Therefore, spatial signaling network data are critical to delineating the net effects of activin and/or TGF-β in the TME.

The identification of cell-specific responses to activin and/or TGF-β requires high-plex technology capable of single-cell resolution, such as flow cytometry; however, these techniques fail to consider the spatial orientation of these cells in situ [20]. The distance between CD8 T-cells and melanoma cells directly influences the efficacy of anti-PD-1 therapy in humans [21]. Novel technologies developed over several years have improved the ability to detect and quantify proteins in situ, including GeoMx digital spatial profiling (DSP), which permits the segmentation of tissue relative to activin co-localization and the subsequent quantification of 50+ proteins [22].

Here, we employed two separate mouse models of CRC with various activin signaling pathway disruptions and performed and DSP analysis on several stage II/III CRC patient samples from the QUASAR II cohort to identify cellular signatures relative to activin co-localization in the TME. Using this approach, we aimed to test the hypothesis that non-canonical activin signaling drives the metastatic behavior of tumor cells while also polarizing immune cells in the TME.

## 2. Materials and Methods

### 2.1. Reagents and Antibodies

Activin was reconstituted in PBS containing 4 mM HCl according to the manufacturer’s instructions (both from R&D, Minneapolis, MN, USA). Inhibitors LY294002 and U0126 were reconstituted in DMSO according to manufacturer’s instructions (both from EMD Millipore Corp. Burlington, MA, USA).

### 2.2. Immunohistochemistry

Immunohistochemical staining (IHC) of α-SMA, pAKT, pERK (all cell signaling, Danvers, MA, USA), TGFβ1, CD4 (both from Abcam, Waltham, MA, USA), and activin (Inhibin-βA, Ansh Labs, Webster, TX, USA) was performed as previously described [23]. In all cases, slides were blindly scored by 2 independent investigators and received scores of 0 (no staining), 1 (intermediate staining), or 2 (high staining). IHC staining and scoring was relied upon to generate the data displayed in Figures 1A–C and 2A,B.

### 2.3. Ts4-Cre; cApc^flox^; Smad4^flox^ Mouse Model (TcAS)

Ts4-Cre; cApc^flox^ C57BL/6J mice were provided by Dr. Khazaie, while at the Robert H. Lurie Comprehensive Cancer Center, Northwestern University, Chicago, IL. Smad4 floxed Smad^4tm2.1Cxd^ were purchased (Jackson Labs, Bar Harbor, ME, USA) and crossed with Ts4Cre mice, and mice harboring the loxP flanked the Apc gene individually. The model was established as follows: to imitate the classical pathway of colon carcinogenesis, loxP sites were introduced with a homologous recombination to exon 11 and 12 of the adenomatous polyposis coli (Apc^lox468^) gene as described in [24]. The Cre gene was inserted to cells that express fatty-acid-binding protein1 (Fabp1, also known as TS4), which is predominately expressed by epithelial cells in the colon with some in the terminal ileum [25]. Recombination by Cre leads to a truncated protein at exon 468 in the Apc gene. The resulting Ts4Cre/Smad4^fl/+^ and cApc^fl/fl^/Smad4^fl/+^ mice were bred to establish the following genotypes: Ts4Cre/cApc^fl/+^/Smad4^+/+^, Ts4Cre/cApc^fl/+^/Smad4^fl/+^, and Ts4Cre/cApc^fl/+^/Smad4^fl/fl^. To generate greater numbers of the latter genotype cohort, Ts4Cre/cApc^fl/+^/Smad4^fl^/^+^ mice were crossed to Ts4Cre/cApc^fl/+^/Smad4^fl/fl^ in a manner that maintained heterozygous Cre and Apc (cApc^fl/+^). This approach necessitated a breeding scheme that would independently produce Ts4Cre/cApc^fl/+^/Smad4^+/+^ mice (see Appendix A). Mice were bred and kept at the animal facility of the University of Illinois in Chicago (UIC), under ACC protocol with specific pathogen-free conditions and had health checks at least once per week. A schematic of this genetic approach, the genotyping data, and the PCR primer information are provided in Appendix A.

### 2.4. Patient Cohort Tissue Microarray

The QUASAR 2 study was an international, multicenter, open-label, phase 3, randomized, controlled trial [26]. This clinical trial was conducted in accordance with the protocols Good Clinical Practice, European Directives 2001/20/EC and 2005/28/EC, and the Declaration of Helsinki. Study approval was obtained from the West Midlands Research Ethics Committee (Edgbaston, Birmingham, UK; REC reference: 04/MRE/11/18). Recruitment consisted of CRC patients ≥ 18 years old and occurred between 2005 and 2010 from 170 hospitals across 7 countries. Briefly, 1941 histologically proven high-risk stage II and III R0 CRC patients, who had sufficient clinical data to determine the respective risk group and had known cancer-specific survival status, consented to the use of tissue samples and did not withdraw this consent at any time that they were included in the study [27]. Patients were randomly assigned to be treated with capecitabine or capecitabine plus bevacizumab. Disease-free and overall survival at 3 years did not differ between the groups. The trial results were published in the *Lancet* [26] and disseminated to patient groups via relevant CRC patient support groups. Tissue samples employed in our study had been collected prior to any treatment of any kind (i.e., radiation, chemotherapy, or clinical trial compounds). Collected tissue samples included a CRC tissue resection obtained at the time of curative-intent surgery, which was formalin-fixed upon collection. From these samples, several formalin-fixed, paraffin-embedded TMAs were generated containing 1.0 mm^3^ CRC tissue punches. We obtained serial sections from 10 blocks which included 1055 cases from the trial which were stained for activin (n = 1042), TGF-β (n = 971), and CD4 (n = 990). Slides were scored according to intensity of the stain. One TMA slide containing 116 patient samples was used for DSP analysis (see below).

### 2.5. Digital Spatial Profiling (DSP, NanoString)

The DSP assay (NanoString Technologies, Seattle, WA, USA) was performed as previously described [28]. Following tissue rehydration and antigen retrieval, the slide was stained with the 4 morphology markers and 57 quantifiable antibodies (all NanoString Technologies, Seattle, WA, USA). The activin morphology marker was custom conjugated to AF647 using the Alexa Fluor 647 Antibody Labeling Kit (ThermoFisher Scientific, Waltham, MA, USA). The quantifiable antibodies are conjugated to oligonucleotide “barcodes” via a photocleavable linker. The resulting image permits visualization of the four fluorescent morphology markers, which can be used to strategize UV light exposure and collection of the quantifiable barcodes. The tumor cell marker PanCK and the pan-immune cell marker CD45 were used to strategize the region of interest (ROI) selection to obtain several ROIs that were considered tumor-dominant or stroma-dominant. These ROIs were then further separated into activin (+) and activin (−) compartments based upon expression of the fluorescent activin morphology marker. UV light was applied to the activin (+) compartment to permit quantifiable barcode release and collection. This process was then repeated in activin (−) compartments. A total of 27 patient samples were used to select 28 ROIs which were separated into 56 AOIs (28 activin (+) and 28 activin (−)) for collection and analysis. The collected barcodes were then hybridized and quantified using the nCounter Sprint Profiler (NanoString Technologies, Seattle, WA, USA), as previously described [29]. Data QC was performed to remove samples with low binding density (0.1–1.8 spots/µm^2^), low nuclei count (<20), abnormal synthetic positive control spike-in expression (<0.3 or >3.0), low surface area (<1600 µm^2^), and low field-of-view detection (<75%) in the nCounter Sprint Profiler, as recommended by NanoString. Background correction was then performed on the samples using a signal:background ratio of counts-of-each-target:negative-controls ratio (negative controls: Ms IgG1, Ms IgG2a, and Rb IgGl). All samples had a signal:background ratio of <3 suggesting they were well within the detection limit of the assay [30]. The geometric mean expression of the housekeeping proteins S6 and GAPDH were used for data normalization based upon the correlation of their expression across all 56 samples, which passed QC. Data were expressed as normalized counts. The DSP technology was relied upon to generate the data in Figures 3–5 and 6A–C.

### 2.6. Transwell Migration Assay

A transwell migration assay was performed as previously described [8]. Briefly, Fibronectin (2 μg/mL; Sigma-Aldrich, St. Louis, MO, USA) was used as a chemoattractant and 2 × 10^5^ cells per well were permitted to migrate for 6 h following stimulation. The migrated cells were stained with DAPI, imaged on an Evos FL Auto 2D, and five microscopic fields of each well were taken and manually counted (ThermoFisher Scientific, Waltham, MA, USA). Each n-value represents the average count of one transwell. Experiments were performed in triplicate or quadruplicate and were repeated six times to ensure that the observed effects were conserved across passages.

### 2.7. Subcutaneous Tumor Model

The CT26 mouse colon cancer cell line (gift from Ajay Maker, University of San Francisco, CA, USA) were stable transfected with a control vector piLenti-siRNA-GFP or with siRNA for activin in piLenti-siRNA-GFP with puromycin resistance (Applied Biological Materials, Richmond, BC, Canada). Then 1 × 10^6^ CT26 CRC cells were injected into the right flank of female BALB/c mice, and tumor growth was measured using a skinfold caliper (Fisher Scientific, Hampton, NH, USA) at days 0, 8, 11, 15, and 23 post-inoculation. The tumor volume was calculated using the modified ellipsoidal formula: $\frac{1}{2}(Length\times {Width}^{2})$, as previously described [31].

### 2.8. Statistical Analysis

Survival data in Figure 1 were analyzed via Log-rank (Mantel–Cox) test; the scoring data and tumor cell migration data were analyzed via ordinary, one-way ANOVA with Tukey multiple comparisons test (GraphPad Prism 9 software, * *p* < 0.05, ** *p* < 0.01). The effect of CD4 on survival using a cox proportional hazards model controlling for age (linear, quadratic and cubic trends), sex, MSI, and cancer stage (II or III) was assessed for the data displayed in Figure 2. The effect of activin and TGF-β using a second cox proportional hazards model, including the interaction of activin and TGF-β and the same covariates as above, was also analyzed for the data displayed in Figure 2. All TMA survival analyses were performed in R version 4.2.1 in consultation with M. Nash, a biostatistician, at the Department of Biostatistics, University of Washington. Linear mixed modeling (LMM) was performed on all DSP data with the Benjamin–Hochberg correction test (NanoString DSP Analysis Suite Software version 2.5.1.145). Data obtained from the LMM analysis were used to generate the heatmap included in Figure 4 using Rstudio 2022.07.02 and the package “gplots” (* *p* < 0.05, ** *p* < 0.01, *** *p* < 0.001, **** *p* < 0.0001). Tumor growth data in Figure 6 were analyzed via ordinary, two-way ANOVA with Sidak multiple comparisons test and AUC data analyzed via ordinary, two-tailed student’s t-test (GraphPad Prism 9 software, * *p* < 0.05, **** *p* < 0.0001).

## 3. Results

### 3.1. Loss of SMAD4 in Murine Intestinal Tumors Is Associated with Increased Activin, pAKT, and CD4 and Decreased Survival

We recently reported that activin leads to migration and metastasis in CRC via non-canonical Smad4-independent signaling [32]. We also reported that activin treatment of CRC cells increased pAKT activation in the absence of SMAD4 in vitro [8]; therefore, we aimed to test the effect of activin-induced PI3K/AKT activity in CRC in vivo. To achieve this, we employed a genetic CRC mouse model with tissue-specific deletion of the canonical SMAD4 signaling protein in the intestinal epithelial cells to determine how removal of this pathway alters colonic activation of the PI3K/AKT pathway and survival in vivo. Histopathological analysis of the colon from the Smad4 wild type (WT) mice (Smad4 +/+), Smad4 heterozygote (Smad4+/−) mice, and Smad4 knockout (KO) mice (Smad4−/−) revealed that expression of activin was significantly upregulated in KO mice (1.16 ± 0.21 scoring units, n = 4) when compared to Smad4+/− mice (0.45 ± 0.11 scoring units, n = 12) with no significant difference in WT mice (0.63 ± 0.13 scoring units) (Figure 1A). Similarly, KO mice displayed a significant increase in pAKT expression (1.58 ± 0.18 scoring units, n = 4) when compared to Smad4+/− mice (0.72 ± 0.14 scoring units, n = 12), with no significant difference across WT mice (0.90 ± 0.19 scoring units, n = 7) (Figure 1B). Interestingly, CD4 expression was increased in both Smad4−/− (1.25 ± 0.27 scoring units, n = 4) and Smad4+/− mice (1.09 ± 0.11 scoring units, n = 12) when compared to Smad4+/+ mice (0.17 ± 0.08 scoring units, n = 7), suggesting that the effects of Smad4 removal may result in discrete changes in T-cell phenotypes that cannot be identified via CD4 IHC (Figure 1C). Representative IHC images can be found in Appendix A.

In a survival analysis (log-rank test, *p* = 0.0018), Smad4+/+ showed significantly longer survival (8.09 ± 1.05 months, n = 7) than Smad4+/− mice (6.15 ± 3 months, n = 12). Smad4−/− mice had significantly shorter survival (3.8 ± 0.4 months, n = 4) than the other 2 groups, which had a similar survival (Figure 1D). Additionally, WT mice did not develop extensive polyposis. In 38% of the WT mice, we found small, macroscopic polyps, and 13% developed large polyps. The animals showed no signs of bowel obstruction. In the Smad4+/− group, 64% had remarkable polyposis in the colon, 45% had signs of bowel obstruction, and 45% had bloody feces, consistent with large polyps. In KO mice, 50% had bowel obstruction and 50% had bloody stool, due to worsening conditions, compared to the other two groups. This supports that increased activin is signaling through the PI3K/pAKT pathway and is associated with outcome in mice lacking Smad4.

### 3.2. Increased Activin Expression Is Required for TGF-β-Associated Improvements in Patient Outcome in Stage II or III CRC

Activin signaling is required for the TGFβ stimulation of metastasis of CRC cells, indicating significant cross-talk in the tumor microenvironment (TME) [4,33]. Thus, we performed IHC staining and scoring for activin, TGF-β, and CD4 on 1055 stage II/III CRC patient samples from the QUASAR II cohort to correlate changes in survival to pan-tumoral expression of these markers. Histopathological analysis of 990 CRC patient tumor samples revealed increased overall CD4 expression in the TMA, which was associated with a significant increase in survival time (chisq(1) = 7.2973, *p* = 0.0069). Additionally, higher CD4 expression in the tissue was associated with lower hazard of mortality (HR = 0.5326, 95% CI from 0.3343 to 0.8483, comparing CD4 = 0 to CD4 = 2, *p* = 0.0080). Estimated probability of survival at 5 years was 71.84% for low CD4 (i.e., CD4 = 0, 95% CI from 65.10% to 79.28%) and 83.85% for high CD4 (i.e., CD4 = 2, 95% CI from 78.34% to 89.74%) (Figure 2A). Activin (n = 1042) and TGF-β (n = 971) together significantly predicted survival (chisq(3) = 9.687, *p* = 0.0214). The combination of low activin (activin = 0) and low TGF-β (TGF-β = 0) was associated with the lowest estimated probability of survival at 5 years (60.86%, 95% CI from 47.74% to 77.57%). This was significantly lower than the combinations of high activin (activin = 2) and high TGF-β (TGF-β = 2, estimated probability of survival 86.73%, 95% CI from 78.43 to 95.91, *p* = 0.0054), high activin and low TGF-β (82.87%, 95% CI from 76.14% to 90.18%, *p* = 0.0031), or low activin and high TGF-β (86.74%, 95% CI from 78.43% to 95.92%, *p* = 0.0035). No other pair of combinations significantly differed from each other (Figure 2B). Taken together, a lack of expression in both ligands conferred the worst outcome, with one ligand alone being able to rescue this effect. Therefore, counter to our initial hypothesis, in CRCs without distant metastasis, low activin in combination with low TGF-β was associated lower survival, while patients with tumors with high activin and low TGF-β fared significantly better. Representative IHC images can be found in Appendix A.

### 3.3. Activin Co-Localizes with Both Tumor and Immune Cells in the TME to Generate Activin-Dependent Signaling Compartments

To further determine the cellular signatures and PI3K/AKT activation status relative to activin co-localization in the TME, we performed DSP analysis on 27 separate patient samples with activin included as a fluorescent marker to permit the quantification of 57 proteins of interest relative to activin co-localization. We specifically included several immune signaling modules and the PI3K/AKT signaling module provided by NanoString in our DSP analysis to identify the activation status of immune cells and the PI3K/AKT pathway in the presence/absence of activin in the TME. This method generated images of our TMA with our four fluorescent markers (Figure 3). Regions of interest (ROIs) were selected throughout the TMA, and the PanCK antibody was used to distinguish tumoral vs. stromal compartments (Figure 3A,B,E,F). These ROIs were then further separated into activin (+) (pink) and activin (−) (gray) areas of illumination (AOIs) (Figure 3C,D,G,H). In summary, this technology permits the quantification of several proteins in situ to confirm changes in protein expression relative to specific tissue compartments (tumoral vs. stromal) and relative to activin co-localization.

Activin plays a direct role in T-cell stimulation in the context of lung cancer through ACVR1 [15]. To further explore the role for activin signaling in the TME and to identify if any easily identifiable trends were observed across activin (+) and (−) AOIs across tumoral and stromal compartments, a heatmap was generated and separated into 4 parts (Figure 3I–L, n = 8, 10, 18, and 20). The heat signature was increased in the activin (+) AOIs in both the tumoral and stromal compartments indicating an increase in proteins implicated in T-cell exhaustion (Figure 3I) and PI3K/AKT signaling (Figure 3J). Markers of stromal tissue (ɑ-SMA) and immune cells (CD45, CD4, CD8) were elevated in the samples labeled as such, validating the separation of samples and the quantitative method (Figure 3K). Furthermore, PanCK had the highest heat signature in the tumor samples independent of activin co-localization, further validating the data set (Figure 3L). Interestingly, activin (−) stromal AOIs had the highest heat signature for HLA-DR (MHCII), which is found on antigen-presenting cells (APCs); however, proteins that are implicated in APC activation were greatest in activin (+) AOIs (Figure 3L). These data suggest that tissue compartments were successfully separated and that activin polarizes immune cell signatures through the PI3K pathway.

### 3.4. Markers of Tumoral and Stromal Compartments Confirm Successful Segregation of Regions with an Increase in CD45 Observed in Activin (−) Regions of the Stroma

In order to validate our DSP approach, we first analyzed the data comparing the tumoral and stromal compartments to confirm that proteins expressed by cells found in the stroma were highest in that compartment and vice versa. Volcano plots revealed PanCK, ɑSMA, and CD45 were the most significantly differentially expressed proteins across the tumor and stroma, which further validated the qualitative separation of the samples, the method of quantification, and the sample integrity (Figure 4A). Given the recent evidence that activin has profound effects on inflammatory cell communication and activation in the setting of lung cancer [15], we next analyzed data in the stroma across activin expression to determine if activin was polarizing the immune cell signatures. Additionally, several more proteins were upregulated in activin (+) AOIs in the stroma (Figure 4B). Next, we generated histograms of several of the significantly differentially expressed proteins in both the tumoral and stromal compartments to determine what signatures were associated with activin co-localization in each compartment. The expression of ɑSMA was significantly higher in the stroma compared to the tumor, and activin (−) AOIs had higher expression of ɑSMA than activin (+) AOIs in the stroma (Figure 4C). Conversely, PanCK expression was significantly upregulated in the tumor samples when compared to the stroma samples and was highest in the tumoral activin (+) AOIs (Figure 4D). Activin (−) AOIs in the stroma had greater CD45 expression than stromal activin (+) AOIs (Figure 4E). A significant increase in both CD4 and CD8 were observed in the stromal compartment of CRC tissue samples regardless of activin co-localization (Figure 4F,G). Therefore, the separation of tumoral and stromal compartments was successful, and activin co-localization is associated with a reduction in immune cell number, which is not attributed to T-cells. A summary of the results can be found in Appendix A.

### 3.5. Activin Stimulates T-Cell Suppression and APC Activation in the Stromal Compartment of the TME in CRC

There has been conflicting evidence surrounding the effect of activin on T-cell polarization [14,15]; however, activin appears to promote macrophage activation [12,13], suggesting a critical role for this cytokine in inflamed tissue. Therefore, several markers of T-cell and APC activation status were included in our panel of quantified proteins (see Appendix A) to determine how activin influences immune cell phenotyping in the TME. The normalized data revealed that the activin (+) compartment displayed a significant increase in several markers of T-cell exhaustion including CTLA-4, FOXP3, and CD25 (Figure 5A–C). Additionally, an increase in PD-1 was only observed in activin (+) tumor samples when compared to activin (−) tumor samples with no significant changes observed in the stroma (Figure 5D). Interestingly, these effects were coupled with an increase in markers of APC activation including CD80 and CD40 (Figure 5D,E). This may result in an increase in APC activity in activin (+) AOIs; however, this is dependent upon the T-cell antigen it interacts with (i.e., CTLA-4 vs. CD28). These data suggest activin co-localization stimulates T-cell exhaustion and expression of activation markers on APCs in both the tumoral and stromal compartments of the TME. A summary of the results can be found in Appendix A.

### 3.6. Activin Co-Localization Is Associated with Increased Activation of PI3K/AKT Signaling to Enhance Tumor Cell Migration and Growth

Our group has previously identified that activin preferentially signals along the PI3K/AKT pathway in the absence of SMAD4 while TGF-β favors the MAPK/ERK pathway in the absence of SMAD4 in CRC [8]. Therefore, we included both the PI3K/AKT and MAPK/ERK protein panels in our DSP analysis (see Appendix A). Several proteins in the PI3K pathway were increased in the activin (+) AOIs independent of the tissue compartment, including PLCG1, Phospho-PRAS40, and Phospho-Tuberin (Figure 6A–C). A summary of the DSP results can be found in the Appendix A.

Activin stimulates tumor cell migration; however, it is unclear which receptor and pathway are responsible for this outcome [32]. Given the observed increase in proteins found in the PI3K/AKT signaling pathway in activin (+) regions of the tumor, we next explored the potential role for activin and the PI3K pathway in transwell migration assays performed on the ACVR2A-expressing HCT116+chr2 cell line. This cell line displayed an increase in migratory ability when pre-treated with activin (18.93 ± 3.46 cells, n = 20) and when compared to control-treated cells (2.24 ± 0.41 cells, n = 18). Activin-stimulated migration was ablated in the presence of the PI3K inhibitor LY294002 (4.35 ± 0.86 cells, n = 20). This effect was not observed when cells received both activin and the MAPK inhibitor U0126 (21.58 ± 5.68 cells, n = 20). Furthermore, the addition of LY294002 alone (6.53 ± 1.78 cells, n = 20) or U0126 alone (7.29 ± 2.12 cells, n = 19) did not alter the migration rate of the CRC cells (Figure 6D).

Activin has been well established as a critical mediator of epithelial to mesenchymal transition [35]; however, activin has also been identified to promote tumor cell proliferation [36], suggesting a potential role in tumor establishment and growth. To study the role for activin in the context of tumor establishment and growth in vivo, we knocked down activin production in CT26 CRC cells, injected them subcutaneously into mice, and measured tumor growth intermittently (see Appendix A). Validation data of the activin KD in CT26 CRC cells can be found in Appendix A. At 23 days post-inoculation, tumor growth was significantly reduced in mice that received the activin KD cells (839.10 ± 291.07 mm^3^, n = 5) when compared to mice injected with control CT26 cells (3510.30 ± 543.77 mm^3^, n = 5) (Figure 6E). Area under the curve (AUC) analysis revealed a significant reduction in total area in the mice receiving activin KD cells (5765 ± 2666 AUC) when compared to mice who received control CT26 cells (20,601 ± 5284 AUC), suggesting a reduction in tumor establishment and growth throughout the experiment (Figure 6F). Interestingly, no histological changes in the tumor were observed across these conditions (see Appendix A). In summary, activin co-localization is associated with an increase in activated-PI3K protein expression which stimulates tumor cell migration in vitro. Additionally, autocrine activin signaling is essential to tumor establishment and growth in vivo.

## 4. Discussion

### 4.1. SMAD-Independent Signaling in CRC

Inactivating mutations in the downstream canonical SMAD pathway are found in 30% of CRC at later stages [6]. Several mouse models of CRC have been generated which leverage genetic models (i.e., *Apc^Min+/^*^−^) and/or chemical agents (i.e., azoxymethane), all of which have been limited by several off-target effects, inconsistent tumor generation, and tumors in regions outside of the colon [37]. To test the effects of activin signaling through the canonical SMAD pathway in the setting of CRC, we employed an epithelial cell-specific SMAD KO mouse Apc model [24,25]. This model provided significantly fewer off-target effects than systemic methods and alternative genetic targets (i.e., *Villin-CreERT2*), which generate polyps in the small intestine [37]. Our data further support the notion that non-canonical activin (Figure 1A) signaling through pAKT (Figure 1B) supports cancer growth and invasion in vivo (Figure 1D) and expands on our previous work that non-canonical activin utilizes distinct mitogenic signaling pathways that affect metastasis [38].

### 4.2. TGF-β, Activin, and CD4 in Human CRC

We previously have shown that TGF-β exerts a pro-metastatic effect in CRC, which is activin-dependent [4]. We interrogated 1055 stage II and III CRC patient samples from a well-validated, robust cohort for IHC scoring of CD4, activin, and TGF-β. The fact that high CD4 correlated with better outcome (Figure 2A) validates the cohort, tissue samples, and IHC scoring since previously reported data from our cohort and others have shown improved outcomes associated with high CD4 expression in the TME [39]. No correlation between activin and CD4 expression was observed, further supporting the hypothesis that activin promotes immune cell phenotyping beyond CD4. Low activin as well as low TGF-β correlate with the worst outcome and either ligand can rescue this effect if robustly expressed (Figure 2B). Neither activin nor TGF-β showed this effect independently, which further supports the need to assess signaling pathways in a broader context. These findings directly contradict our previously published work [4] showing low activin and low TGF-β are associated with improved outcomes in CRC; however, this data set provides a more robust cohort across multiple stages of CRC, providing a more reliable insight into the role for these molecules in the TME and their influence on survival. We have also previously reported that activin production is significantly altered in stage IV CRC specifically; therefore, future studies should include stage IV patients in the quantification of activin and TGF-β, as these patients may have the greatest level of dysregulation [33].

This data provides valuable insight into the combined contribution of TGF-β and activin expression in the TME on the outcomes of stage II and III CRCs. We were intrigued by these somewhat counterintuitive yet robustly significant findings and set out to understand underlying regional and cell-specific effects with the understanding that they may not be generalizable.

### 4.3. Activin Co-Localization with Fibroblasts and Immune Cells

In order to maximize the information gained from our patient samples, we specifically chose to employ DSP technology, which permits the quantification of several proteins (>50) and avoids wasting several slices of the TMAs on iterative IHC stains for individual proteins of interest. Cancer associated fibroblasts (CAFs) secrete increased amounts of activin compared to quiescent fibroblasts, providing a significant source of activin in the stromal compartment of the TME [4,40]. Interestingly, DSP data identified that expression of the CAF marker ɑSMA was significantly increased in the activin (−) areas of the stroma (Figure 4C). Previous work has identified that activin stimulates the expression of ɑSMA; however, this appears to be mediated through the SMAD signaling pathway [41]. Stromal activin (+) AOIs displayed a significant increase in several proteins found in the PI3K pathway when compared to stromal activin (−) AOIs, suggesting a potential switch to non-canonical activin signaling in the stroma which stimulates alternative outcomes in CAFs (Figure 6A–C).

In our cohort, we observed a significant increase in CD45 in stromal activin (−) AOIs when compared to activin (+) AOIs, suggesting an increase in the total number of immune cells in activin (−) AOIs (Figure 4E). However, no change in amounts of CD4 or CD8 were identified across stromal activin compartments, indicating that T-cells were not responsible for the increase in CD45 expression (Figure 4F,G). In line with previously reported data that activin stimulates CD80 expression on macrophages in vitro [42], increases in the co-stimulatory molecule CD80 were observed in activin (+) AOIs (Figure 5E). Interaction of CD80 on APCs with CD28 on CD4 T-cells is required for the co-stimulation of the CD4 T-cells [43]. However, it was recently demonstrated that CD80 has a greater affinity for CTLA-4, an immune checkpoint which inhibits T-cell functions, when compared to CD28 [44]. Therefore, increased expression of CD80 in the TME coupled with increases in CTLA-4 are likely to have a net-immunosuppressive effect. Indeed, the expression of CTLA-4 on T-cells interacts with CD80 or CD86 to reduce proliferation, cytokine production, and cytotoxic responses [45]. T-cells, which are CD8+CD25+Foxp3+, have been confirmed to be regulatory, specifically in the setting of CRC [46]. Therefore, the increased expression of these markers where activin co-localized (Figure 5A–C) suggests a potent immunoregulatory role for activin in CRC.

Previous work has identified that canonical activin signaling is disrupted in both MSI and MSS CRC [47,48], suggesting that Smad-independent signaling is present in the TME regardless of MSI status. Additionally, activin preferentially signals along the PI3K/AKT pathway in the absence of SMAD4 in CRC cells [8]. In line with this previously reported data, we observed increases in several proteins found in the PI3K/AKT pathway, including PLCG1, Phospho-PRAS40, and Phospho-Tuberin (Figure 6A–C). Taken together, these data suggest that non-canonical activin signaling through the PI3K/AKT pathway promotes immunosuppression in the TME.

### 4.4. Activin Signaling in the Tumor Cells

Tumor cells produce significant amounts of activin, and this production contributes to the signaling patterns observed in the TME [49]. Our DSP data confirmed that activin (+) AOIs contain the greatest amount of PanCK, suggesting that activin co-localized with tumor cells which were likely producing significant amounts of activin (Figure 4D). The activin produced by these tumor cells can then signal in an autocrine/paracrine manner to enhance pathway activation in these cells. Several proteins in the PI3K/AKT pathway were found to be upregulated in activin (+) AOIs in the tumoral compartment, including PLCG1, Phospho-PRAS40, and Phospho-Tuberin, suggesting a possible switch to non-canonical PI3K/AKT signaling in tumor cells (Figure 6A–C). Previous work has identified that activin stimulates the migration of cancer cells in an SMAD-independent manner [8]. Here, we found that activin stimulates the migration of CRC cells, which is ablated in the presence of the PI3K inhibitor (LY) (Figure 6D), suggesting that activin stimulation of the PI3K/AKT pathway is required for activin-mediated tumor cell migration. Our data to this point suggests a potential role for non-canonical activin signaling in tumor tolerance (immunosuppression) and tumor cell migration; however, recent evidence suggests that activin also plays an important role in supporting tumor growth [36]. We explored the potential for activin in tumor establishment and growth in vivo via CT26 cells with/without siRNA for *INHBA*. We found that the removal of tumoral-derived activin inhibits the establishment and growth of tumor cells in vivo (Figure 6E). These data provide compelling evidence that activin stimulated non-canonical PI3K signaling in the TME drives tumor growth and metastasis in situ, in vitro, and in vivo.

## 5. Conclusions

These data provide evidence that activin (i) signals non-canonically through the PI3K/AKT pathway and is associated with reduced survival time in a mouse model of CRC, (ii) is necessary for TGF-β associated improved outcomes in CRC (iii) on a local level, is associated with increased markers of T-cell exhaustion in both the tumoral and stromal compartments of the TME, and drives tumor growth in vivo and migration in vitro. Surprisingly, IHC data identified that increased activin in CRC was associated with improved outcomes, which was interdependent with TGF-β, while DSP data suggested that activin stimulates T-cell exhaustion. Activin signaling is highly context-dependent and cell-specific, which may mediate several opposing outcomes at the organismal level. This seemingly contradictory data further underscores the need to assess targetable signaling networks on various planes to fully understand the various roles that come together to support tumor growth and spread. Furthermore, the application of cutting edge, commercially available technology such as DSP to identify signaling networks provides significantly more insight into the state of the TME when compared to IHC and, therefore, may provide more reliable information regarding the potential success of each therapeutically available option.

## Figures and Tables

**Figure 1 cancers-15-03003-f001:**
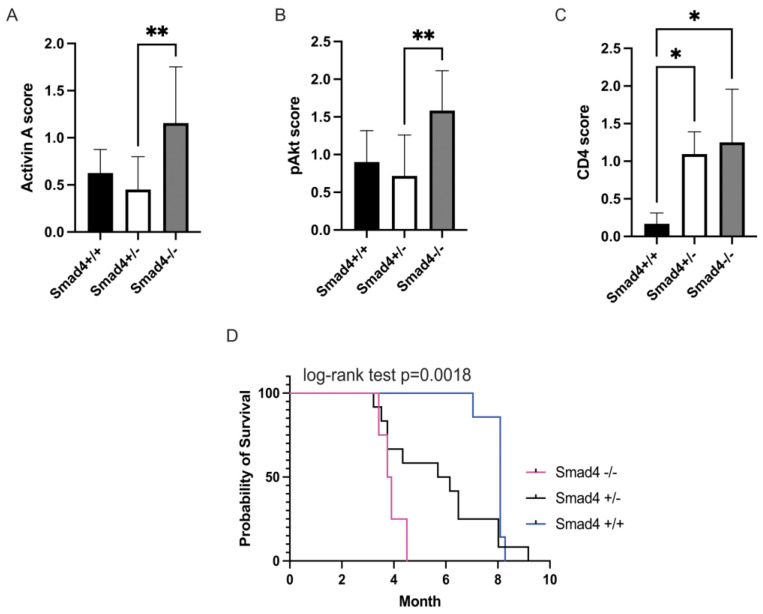
Ablation of Smad4 in APC mice leads to decreased survival and is associated with increased activin, pAKT, and CD4. Colon tumors of Smad4−/− mice show significantly increased (**A**) activin, (**B**) pAKT, and (**C**) CD4 compared to Smad4−/+ and Smad4+/+ mice using histopathology. (**D**) Kaplan–Meier survival curves of Ts4Cre/cApcfl/+/Smad4+/+ (WT) (blue), Ts4Cre/cApcfl/+/Smad4+/− (het) (black), and Ts4Cre/cApcfl/+/Smad4−/− (KO) (magenta) show the loss of survival with ablation of Smad4 (log-rank test *p* = 0.0018), a downstream signaling molecule of canonical SMAD signaling. Data analyzed via ordinary, one-way ANOVA with Tukey multiple comparisons test (**A**–**C**) or Log-rank (Mantel–Cox) test (**D**) (* *p* < 0.05, ** *p* < 0.01; n = 4, 7, and 12).

**Figure 2 cancers-15-03003-f002:**
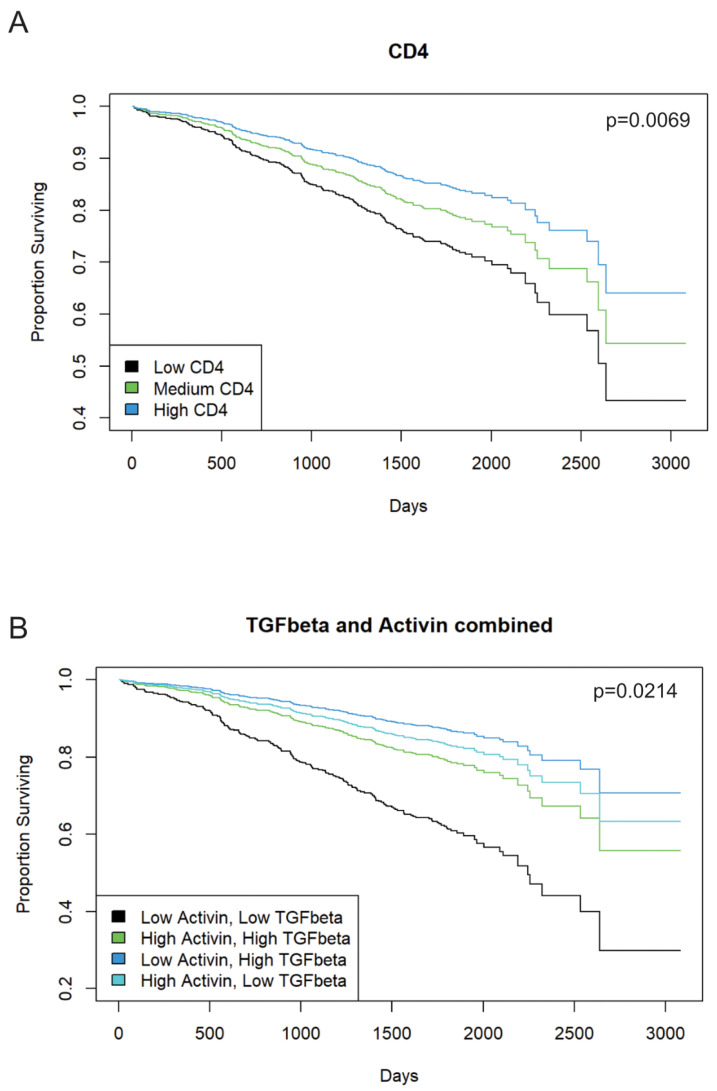
Pan tumoral activin or TGF-β expression in IHC staining is associated with improved patient outcome in stage II or III CRCs. A total of 1055 patients with stage II or stage III CRC from the QUASAR 2 cohort were stained for activin, TGF-β, and CD4 using IHC. (**A**) Confirming the validity of the cohort, tumoral CD4 is associated with better survival (blue; *p* = 0.0069; n = 990). (**B**) Combined low activin and high TGF-β levels are associated with better outcome in CRC patients (dark blue) as was low TGF-β and high activin (light blue). Low activin and low TGF-β are associated with worse outcome (black; *p* = 0.0214; activin n = 1042; TGF-β n = 971).

**Figure 3 cancers-15-03003-f003:**
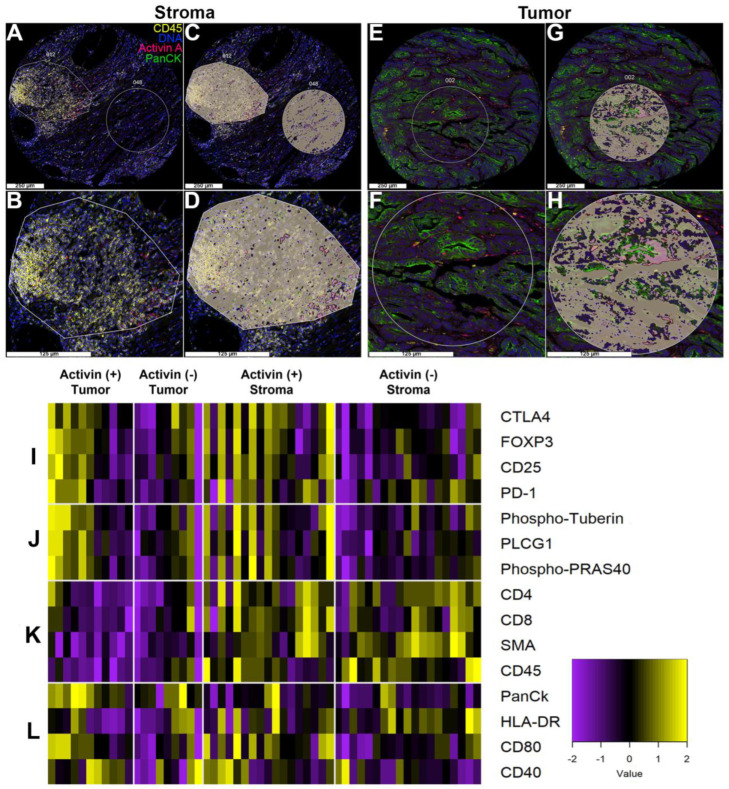
Activin co-localizes with both tumor and immune cells in the TME to generate activin-dependent signaling compartments via DSP. CRC patient tumor samples from the QUASAR 2 clinical trial were stained with four fluorescent markers (PanCK, green; DNA, blue; CD45, yellow; and activin, red) and were separated into stroma or tumor-containing samples using DSP. (**A**,**B**) Representative regions of interest (ROI) in the stroma, (**C**,**D**) which were separated into two distinct areas of illumination (AOI) prior to collection and quantification: activin positive (pink) and activin negative (gray). (**E**,**F**) Representative ROIs in the tumor, (**G**,**H**) separated into activin (+)/(−) AOIs. (**I**) Activin (+) areas of tumor samples in the stroma show the highest heat signature for several markers of T-cell suppression when compared to activin (−) areas. A similar trend was observed in activin (+) areas of the tumor. (**J**) Activin (+) areas in the tumor and stroma have higher levels of PI3K/AKT signaling proteins when compared to activin (−) areas. (**K**) Several markers of the stroma, including immune cell markers, are highest in the stroma independent of activin co-localization. (**L**) The tumor marker PanCK is highest in tumoral compartments, indicating successful classification of tumor vs. stroma. Additionally, several markers of APC activation appeared to be highest in activin (+) areas on the stromal regions. Data are expressed as log2 normalized counts, n = 8, 10, 18, and 20.

**Figure 4 cancers-15-03003-f004:**
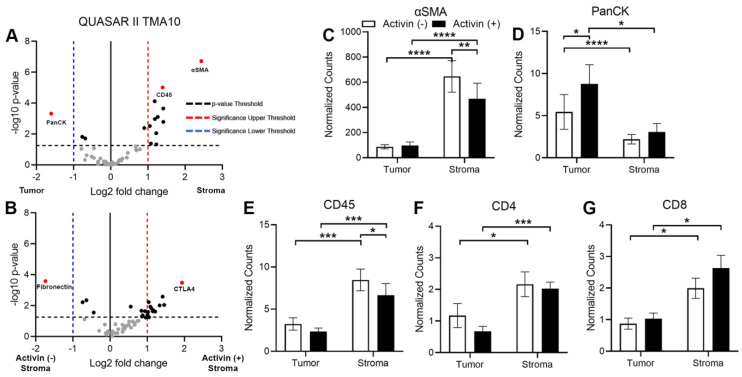
Markers of tumoral and stromal compartments confirm successful segregation of regions via DSP with an increase in CD45 observed in activin (−) regions of the stroma. (**A**) Volcano plot indicating that several proteins, including CD45, α-SMA, and PanCK, are significantly differentially expressed across the tumoral and stromal compartments in the TME. (**B**) Volcano plot displaying several proteins that are significantly differentially expressed in the stromal compartment of the TME, which is dependent upon activin co-localization. Analysis of the CRC tissue sections across activin (+) and (−) areas within tumoral and stromal compartments identified significant changes in expression of several proteins found in the stroma and tumor of the TME, including (**C**) ɑ-SMA, (**D**) PanCK, (**E**) CD45, (**F**) CD4, and (**G**) CD8. Data were analyzed via linear mixed modeling with Benjamini–Hochberg multiple-correction test (* *p* < 0.05, ** *p* < 0.01, *** *p* < 0.001, **** *p* < 0.0001, n = 8, 10, 18, and 20).

**Figure 5 cancers-15-03003-f005:**
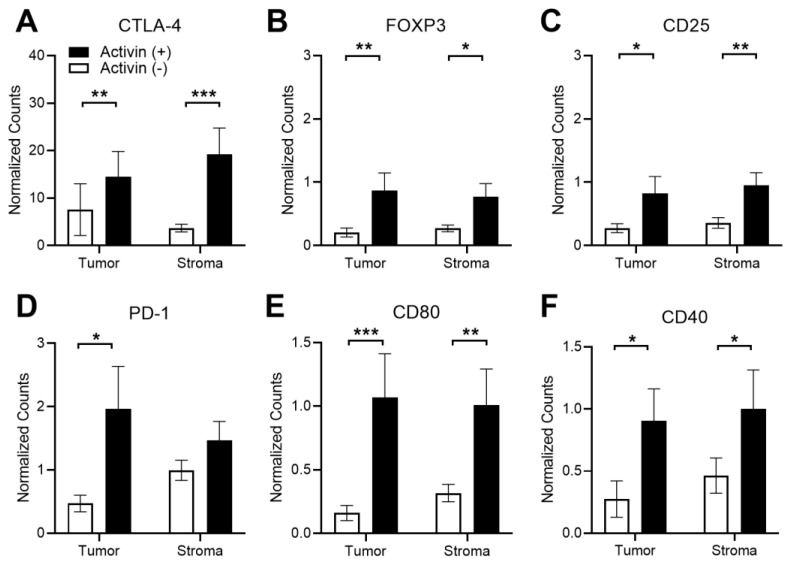
Activin co-localization is associated with T-cell exhaustion and increased APC activation markers in DSP-analyzed images. Several markers of T-cell exhaustion were significantly upregulated in activin (+) areas in both the tumoral and stromal compartment of the TME, including (**A**) CTLA-4, (**B**) FOXP3, and (**C**) CD25. Significant increases in the T-cell exhaustion marker (**D**) PD-1 were only observed in activin (+) areas in the tumor. Additionally, increases in markers of APC activation, (**E**) CD80 and (**F**) CD40, were upregulated in activin (+) AOIs in both tumoral and stromal compartments of the TME. Data were analyzed via linear mixed modeling with Benjamini–Hochberg multiple-correction testing [34] (* *p* < 0.05, ** *p* < 0.01, *** *p* < 0.001, n = 8, 10, 18, and 20).

**Figure 6 cancers-15-03003-f006:**
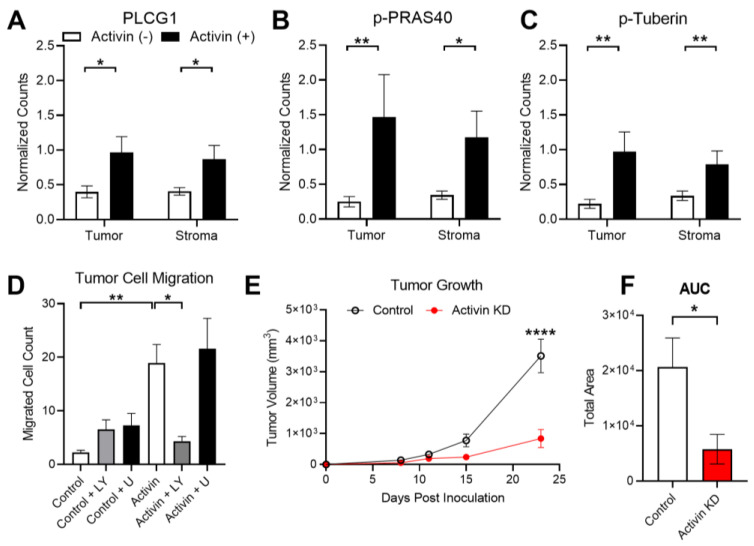
Activin-induced PI3K signaling enhances tumoral growth and metastasis. (**A**–**C**) Activin co-localization was associated with increased expression of several PI3K/AKT markers, including PLCG1, Phospho-PRAS40, and Phospho-Tuberin in the DSP analysis. (**D**) In vitro activin stimulated migration of ACVR2A-expressing HCT116+chr2 CRC cells, which was ablated in the presence of the PI3K inhibitor LY294002 but not the MAPK inhibitor U0126. (**E**) Mice injected with CRC cells which have reduced activin production develop significantly smaller tumors at day 23 post-inoculation. (**F**) Total tumor growth over time is significantly reduced in mice receiving activin KD CRC cells. DSP data analyzed via linear mixed modeling with Benjamini–Hochberg multiple-correction testing; migration data analyzed via ordinary one-way ANOVA with Tukey multiple comparisons test; tumor growth data analyzed via two-way regular ANOVA with effect of time, cell type, and interaction with Sidak’s multiple comparisons test for post-hoc analysis; AUC data analyzed via ordinary two-tailed student’s t-test (* *p* < 0.05, ** *p* < 0.01, **** *p* < 0.0001; (**A**–**C**) n = 8, 10, 18, and 20; (**D**) n = 18, 19, and 20; (**E**,**F**) n = 5).

## Data Availability

All relevant data are included in the manuscript or in the Appendix A.

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
