# Peer review of "Non-Canonical Activin A Signaling Stimulates Context-Dependent and Cellular-Specific Outcomes in CRC to Promote Tumor Cell Migration and Immune Tolerance"

_cancers, 2023, doi:10.3390/cancers15113003_

Round 1
Reviewer 1 Report
The study by Wiley and Bauer et al. shows how active A interacts with the microenvironment of colorectal cancer, thus affecting the patient's outcomes.
The study is well-designed and involves currently applied experimental assays and technologies. It is of high value as the research was conducted on both mouse models and validated with patient samples.
The model revealed that activin A inhibits the activity of immune cells but promotes metastasis in CRC patients, which is a finding of high clinical relevance as activin A is drug-targetable.
Small typos to be fixed.
Author Response
We thank the reviewer for these positive comments.
Reviewer 2 Report
The work done by Wiley et al proposes to address the non canonical role played by Activin in Colorectal cancer. However, the study falls short of what it promises.
The main problem with the manuscript is the lack of focus. The authors seem to address too many aspects which makes the paper confusing despite some really interesting findings.
There is no rationale provided for the experiments done - and in no particular order.
I would recommend a rejection for the manuscript in the present format.
The quality of language used is fine, except for very minor corrections.
Author Response
The main problem with the manuscript is the lack of focus. The authors seem to address too many aspects which makes the paper confusing despite some really interesting findings.
Thank you for the suggestion, we have expanded the introduction, removed some of the statements on the markers of interest in the discussion, and expanded the descriptions of the most impactful findings of our study to improve the focus of the manuscript.
There is no rationale provided for the experiments done - and in no particular order.
We thank the reviewer for noting that we did not provide a clear rationale for the experiments performed and have added statements throughout each of the results sections to provide the rationale for each experiment performed.
*All changes in the manuscript are in red*
Reviewer 3 Report
Mark Wiley and co-authors explore the signaling pathway of activin in CRC tumor cells and tumor microenvironment. This is a well though manuscript that includes both human and murine data. However, the following concerns should be addressed before publication.
Major:
-Number of samples used in Figure 3 I-L. Representative images of the additional markers used in Fig3 I-L should be shown either in the main manuscript or in supplementary data.
-IHC for the activin, PanCk, CD8, CD45 and CD4 graphed on figure 4 should be shown. Representative figures for the quantification and H&E should also be part of the figure. Same comment is applied to figure 5.
-PI3K and MAPK activation (increased p-MAPK, or p-ERK, p38, p-AKT, mTOR) and both canonical and non-canonical pathways should be shown to strength the affirmation that the non-canonical pathways are activated. Additionally, the authors should justify the choice of PLCG1, p-PRAS40 and p-Tuberin. IHC of any of these markers should also been shown, in addition to any of the canonical downstream signals. Moreover, the authors on line 387 state that an increase in PI3K signaling is observed without mentioning the figure/table for this observation.
-The activin expression on cells injected in the mice needs to be shown by western blot, to confirm that activin was KD and remains in the control cells.
-The authors conclude from figure 6 that activin colocalization with PI3K expression stimulates cell migration (line 404) but fail to show the IHC to confirm this in the tumors collected from the mice injected with CT26C cells. Moreover, the authors should show histological samples of both tumors, since I suspect, the histology is going to differ based on the cell migration assay. Would also be interesting to see if the PI3K inhibitor LY294002 reduced tumor growth in the mouse model or if the CT26C cells with activin KD have less migration in the in vitro setting.
Minor:
-Line 37: TMAs abbreviation
-The genotype of all the mice used in breeding and the ones used for survival, should be also provided in supplementary figure 1 (gel figures).
-No indication of the number of mice used in figure 6
-Any time the authors refer to figures and or table on the supplementary date, the number should be sued instead of the general mention that is shown on the supplementary data. For example: Line 341 supplementary table 2, 357, line 368, line 385, and line 398
-In discussion 4.1 when authors refer to data reported the figure associated should be mentioned.
-Line 380 missing reference
n/a
Author Response
Major: Number of samples used in Figure 3 I-L. Representative images of the additional markers used in Fig3 I-L should be shown either in the main manuscript or in supplementary data.
We thank the reviewer for this suggestion. We have added the number of samples used in Figure 3 to both the results and the figure legend to better display this information. We have also added the n-values to Supplemental Table 2 to improve our data transparency.
The representative images used to generate the data in Figure 3 I-L are displayed in Figure 3 A-H. The quantitative markers used to generate the data are not ones that can be visualized. They are antibodies attached to oligonucleotide “barcodes” which are released when exposed to UV light. The four fluorescent markers displayed in the representative images provided in Figure 3 A-H are all that can be visualized on the DSP slide. Regions of Interest (ROIs) were first selected where high levels of activin and CD45/PanCK were observed (Figure 3A-B & 3E-F). Those were then separated into activin (+) and activin (-) areas of illumination (AOIs) based upon activin expression (Figure 3C-D & 3G-H). The activin (+) AOI was first exposed to UV light and the cleaved barcodes were then collected for quantification via nCounter reads. The activin (-) AOI was then exposed to UV light second and those barcodes were quantified separately using nCounter reads. The quantified data from 28 ROIs (56 AOIs) were then analyzed via Linear Mixed Modeling. We have updated figure legend titles and added text to the DSP section of the methods in an attempt to make this information clearer.
-IHC for the activin, PanCk, CD8, CD45 and CD4 graphed on figure 4 should be shown. Representative figures for the quantification and H&E should also be part of the figure. Same comment is applied to figure 5.
We thank the reviewer for suggesting we add more representative images to our manuscript. The data generated for Figures 4, 5, and 6 were all done using the DSP quantitative technology which is capable of quantifying several more proteins than IHC and the data is quantified in a non-subjective manner providing more robust data sets than IHC. The representative images for this data are provided in Figure 3 A-H.
-PI3K and MAPK activation (increased p-MAPK, or p-ERK, p38, p-AKT, mTOR) and both canonical and non-canonical pathways should be shown to strength the affirmation that the non-canonical pathways are activated. Additionally, the authors should justify the choice of PLCG1, p-PRAS40 and p-Tuberin. IHC of any of these markers should also been shown, in addition to any of the canonical downstream signals. Moreover, the authors on line 387 state that an increase in PI3K signaling is observed without mentioning the figure/table for this observation.
We thank the reviewer for the suggestion to provide more data surrounding the ability of activin to stimulate the PI3K pathway. Our lab has previously shown that activin stimulation of CRC cells leads to phosphorylation of Akt which is exacerbated in the absence of SMAD4. This paper also showed that activin stimulation leads to activation of the canonical Smad pathway in CRC cells (PMC4619565). This publication provided our justification for selecting the PI3K/AKT signaling module which includes a panel of proteins found in this pathway and is provided commercially from NanoString for the DSP analysis. This information was added to the introduction to improve the clarity of our rationale.
We specifically selected the DSP technology to allow us to quantify several proteins (>50) on one slide to avoid wasting several slices of our TMAs on iterative IHC stains for each of the markers found in the PI3K/AKT pathway which were included in our analysis. We have added some of this information to the discussion.
It should be made clear that the data for PLCG1, p-PRAS40, and p-Tuberin was obtained via DSP analysis and not IHC, therefore there are no representative images where these markers can be visualized. The representative images for all data obtained from the DSP are included in Figure 3 A-H. We have added information to the figure legend titles and statements to the methods sections to clarify which method was employed to generate the data for each figure. Additionally, we have changed the statement on line 387 from “PI3K signaling proteins…” to “proteins found in the PI3K/AKT signaling pathway…”.
-The activin expression on cells injected in the mice needs to be shown by western blot, to confirm that activin was KD and remains in the control cells.
We thank the reviewer for this suggestion and have added ELISA data to Supplemental Figure 4 from our activin KD cells to provide validation of the KD. Since activin A is a secreted protein, we employed an ELISA to quantify activin A production in our KD.
-The authors conclude from figure 6 that activin colocalization with PI3K expression stimulates cell migration (line 404) but fail to show the IHC to confirm this in the tumors collected from the mice injected with CT26C cells. Moreover, the authors should show histological samples of both tumors, since I suspect, the histology is going to differ based on the cell migration assay. Would also be interesting to see if the PI3K inhibitor LY294002 reduced tumor growth in the mouse model or if the CT26C cells with activin KD have less migration in the in vitro setting.
Thank you for the suggestion, we have added H&E images from the livers of the mice that received the CT26 cells to Supplemental Figure 4 to provide evidence that there were no histological changes identified across the treatments. As shown in the images, tumor burden was very high in both conditions making histological analysis difficult. We performed IHC stains for CD4 and found no difference and did not perform any further histological analysis of these samples. We thank the reviewer for the excellent suggestion to perform the in vivo LY294002 experiment, however the resources are no longer available to obtain the mice and perform the experiment. We also thank the reviewer for the suggestion to perform the transwell migration assay on the CT26 cells, however we exclusively saw changes in tumor growth and establishment in this model. We did not observe any changes is survival rates nor was there any evidence of increased metastasis in the presence/absence of the siRNA for INHBA. Therefore, we did not anticipate any changes in the migratory capacity of these cells, but used the HCT116+chr2 cells instead given that they have a high migratory capacity and that the activin pathway is easily manipulated in these cells.
Minor: Line 37: TMAs abbreviation
Thank you, this change has been made.
-The genotype of all the mice used in breeding and the ones used for survival, should be also provided in supplementary figure 1 (gel figures).
Thank you, this data has been added to Supplemental Figure 1.
-No indication of the number of mice used in figure 6
Thank you, this information has been added to the results and figure legend.
-Any time the authors refer to figures and or table on the supplementary date, the number should be sued instead of the general mention that is shown on the supplementary data. For example: Line 341 supplementary table 2, 357, line 368, line 385, and line 398
Thank you, these changes have been made.
-In discussion 4.1 when authors refer to data reported the figure associated should be mentioned.
Thank you, this has been updated.
-Line 380 missing reference
Thank you, this has been updated
*All changes in the manuscript are in red*
Reviewer 4 Report
Present work by Wiley etal on "non-canonical ...... Immune tolerance" is well written and scientifically sound. There are few comments written below that needs explanation:
In introduction part; a lot of important information related to background literature is missing.
Were the mice genotyped to ensure the proper floxing?
How was IHC data represented? I did not see any representative staining images throughout the manuscript as well as in supplementary material.
In patient cohort tissue microarray section, what was the tissue sample, how was it collected and stored? Patient related information is unclear. The authors included 1941 patients initially, and 1055 were included for staining. There is also a mention of 116 additional patient samples for DSP. All this information needs to be written clearly.
English language is fine. Minor proof reading is required.
Author Response
In introduction part; a lot of important information related to background literature is missing.
Thank you for the suggestion, we have added more background information and relevant literature to our introduction to address this need.
See also comments to reviewer 2 about overall work to improve introduction and rationale.
Were the mice genotyped to ensure the proper floxing?
Thank you for this question, we have added a substantial amount of data to supplemental Figure 1 which now includes gels and primer information for the genotyping of these mice. We have also updated supplemental breeding and genotyping schematic in this Supplemental Figure.
How was IHC data represented? I did not see any representative staining images throughout the manuscript as well as in supplementary material.
Thank you for this suggestion, we have added representative images of the staining and scoring of the IHC images to the Supplemental Figures 2 & 3.
In patient cohort tissue microarray section, what was the tissue sample, how was it collected and stored? Patient related information is unclear. The authors included 1941 patients initially, and 1055 were included for staining. There is also a mention of 116 additional patient samples for DSP. All this information needs to be written clearly.
Thank you for pointing this out, it has been addressed.
*All changes in the manuscript are in red*
Round 2
Reviewer 2 Report
The authors have tried to address the queries by adding more text to the manuscript to help understand the manuscript better.
1. Why was female BALB/C mice used for the invivo study, when the incidence of CRC is higher in males when compared to females?
2. Have the authors checked for the total levels of AKT, to see whether the change occurs because of change in translation or activation?
3. In fig 2, were other cell types (like Tumor Associated Macrophages) which are potent producers of TGFb -and which promote metastasis and tumor growth checked? Any reason why these were omitted?
The language used is fine.
Author Response
*All changes in the manuscript are in blue*
- Why was female BALB/C mice used for thein vivostudy, when the incidence of CRC is higher in males when compared to females?
We specifically chose to employ BALB/c mice given the strong immune response observed in these mice in other models of CRC and the data we obtained suggesting activin is modulating the immune response in the TME of CRC. More specifically, our data and the literature suggested that activin exerts profound effects on T cell function in CRC and BALB/c females show increased T-cell infiltration in models of metastatic CRC. Therefore, we chose to employ these mice to determine if removal of activin had a more potent effect on the model with the greatest T cell inflammatory response.
- Have the authors checked for the total levels of AKT, to see whether the change occurs because of change in translation or activation?
Thank you for this question, we included Pan-AKT in our DSP analysis, but did not observe any statistical differences in expression.
- In fig 2, were other cell types (like Tumor Associated Macrophages) which are potent producers of TGFb -and which promote metastasis and tumor growth checked? Any reason why these were omitted?
We thank the reviewer for bringing up the potential for activin’s influence on other inflammatory cells found in the TME. We did not perform any further IHC stains on our TMAs to avoid wasting several slices from the tissue blocks on iterative IHC staining and scoring. Instead, we chose to perform DSP analysis which permits robust quantification of several markers on one slide. Some of the TAM markers included in our panel were CD68, CD163, and PD-1. No changes in CD68 or CD163 expression were observed, however there was an increase in PD-1 expression in activin (+) AOIs of the tumoral compartment as displayed in Figure 6D. It was not confirmed which cell type was responsible for this increased expression and will be further investigated in our future experiments.
Reviewer 3 Report
The authors addressed the majority of the comments. Small corrections needed:
-Line 80: should be IL1B
-Is there no statistical difference between survival of smad4+/+ mice and smad4-/- mice?
-lines of genotyping gel on supplementary figure 1 should be labeled
- IHC and H&E figures in supplementary information are missing scale bars
-Supplementary table 2: the authors refer that the methods used are described in the supplemental methods, but DSP is described in the methods of the manuscript. Additionally, the table refers to statistical differences, but none is shown of the table, only figure 4-6 show statistical analysis. If other differences were found, they should be noted on the table as they are in the figures.
- On figure 6 D, the vary from 18 to 20, is this number of cells seeded or number of independent experiments done? Not clear. If the experiment was not repeated independently, at least 3 times, it should be.
- if the authors are not showing the H&E sections of tumors in figure 4, should refrain from stating: “no histological changes in the tumor were observed across these conditions” (line 462). And there is no point on showing the liver for a subcutaneous tumor implantation.
n/a
Author Response
*All changes in the manuscript are in blue*
Line 80: should be IL1B
Thank you, this change has been made.
Is there no statistical difference between survival of smad4+/+ mice and smad4-/- mice?
We generated a Kaplan-Meier survival curve and performed a Log-rank Mantel-Cox test which indicated that there was a significant difference in survival across all three groups. The results and methods have been updated to make the information surrounding the analysis performed clearer.
lines of genotyping gel on supplementary figure 1 should be labeled
Thank you, this information has been added.
IHC and H&E figures in supplementary information are missing scale bars
Thank you, this information has been added.
Supplementary table 2: the authors refer that the methods used are described in the supplemental methods, but DSP is described in the methods of the manuscript. Additionally, the table refers to statistical differences, but none is shown of the table, only figure 4-6 show statistical analysis. If other differences were found, they should be noted on the table as they are in the figures.
Thank you, we have removed the mention of supplemental methods. We have also added an additional supplemental table which includes the p-values for each protein of interest from our DSP analysis in an attempt to be more transparent with our data.
On figure 6 D, the vary from 18 to 20, is this number of cells seeded or number of independent experiments done? Not clear. If the experiment was not repeated independently, at least 3 times, it should be.
We thank the reviewer for pointing out that this information needs to be made clearer. The n-value for Figure 6D refers to the number of transwells that were collected for each condition. Each time an experiment was performed, 2 x 105 cells were seeded into each well and each condition was performed in either triplicates or quadruplicates. This experiment was repeated 6 times to ensure that the effect was conserved across passages. We have added some information to the methods section to make this information clearer.
if the authors are not showing the H&E sections of tumors in figure 4, should refrain from stating: “no histological changes in the tumor were observed across these conditions” (line 462). And there is no point on showing the liver for a subcutaneous tumor implantation.
We thank the reviewer for this comment and have removed the statements and supplemental liver images.